# Deleterious Effects of Hyperactivity of the Renin-Angiotensin System and Hypertension on the Course of Chemotherapy-Induced Heart Failure after Doxorubicin Administration: A Study in Ren-2 Transgenic Rat

**DOI:** 10.3390/ijms21249337

**Published:** 2020-12-08

**Authors:** Petr Kala, Hana Bartušková, Jan Piťha, Zdenka Vaňourková, Soňa Kikerlová, Šárka Jíchová, Vojtěch Melenovský, Lenka Hošková, Josef Veselka, Elzbieta Kompanowska-Jezierska, Janusz Sadowski, Olga Gawrys, Hana Maxová, Luděk Červenka

**Affiliations:** 1Department of Cardiology, University Hospital Motol and 2nd Faculty of Medicine, Charles University, 150 06 Prague, Czech Republic; josef.veselka@fnmotol.cz; 2Center for Experimental Medicine, Institute for Clinical and Experimental Medicine, 140 21 Prague, Czech Republic; hana.bartuskova@ikem.cz (H.B.); jan.pitha@ikem.cz (J.P.); zdenka.vanourkova@ikem.cz (Z.V.); sona.kikerlova@ikem.cz (S.K.); sarka.jichova@ikem.cz (Š.J.); olga.gawrys@ikem.cz (O.G.); ludek.cervenka@ikem.cz (L.Č.); 3Department of Cardiology, Institute for Clinical and Experimental Medicine, 140 21 Prague, Czech Republic; vojtech.melenovsky@ikem.cz (V.M.); lenka.hoskova@ikem.cz (L.H.); 4Department of Renal and Body Fluid Physiology, Mossakowski Medical Research Centre, Polish Academy of Sciences, 01-224 Warsaw, Poland; ekompanowska@imdik.pan.pl (E.K.-J.); jsadowski@imdik.pan.pl (J.S.); 5Department of Pathophysiology, 2nd Faculty of Medicine, Charles University, 110 00 Prague, Czech Republic; hana.maxova@lfmotol.cuni.cz

**Keywords:** chemotherapy-induced heart failure, doxorubicin, hypertension, renin-angiotensin-aldosterone system

## Abstract

Doxorubicin’s (DOX) cardiotoxicity contributes to the development of chemotherapy-induced heart failure (HF) and new treatment strategies are in high demand. The aim of the present study was to characterize a DOX-induced model of HF in Ren-2 transgenic rats (TGR), those characterized by hypertension and hyperactivity of the renin-angiotensin-aldosterone system, and to compare the results with normotensive transgene-negative, Hannover Sprague-Dawley (HanSD) rats. DOX was administered for two weeks in a cumulative dose of 15 mg/kg. In HanSD rats DOX administration resulted in the development of an early phase of HF with the dominant symptom of bilateral cardiac atrophy demonstrable two weeks after the last DOX injection. In TGR, DOX caused substantial impairment of systolic function already at the end of the treatment, with further progression observed throughout the experiment. Additionally, two weeks after the termination of DOX treatment, TGR exhibited signs of HF characteristic for the transition stage between the compensated and decompensated phases of HF. In conclusion, we suggest that DOX-induced HF in TGR is a suitable model to study the pathophysiological aspects of chemotherapy-induced HF and to evaluate novel therapeutic strategies to combat this form of HF, which are urgently needed.

## 1. Introduction

The past two decades have brought a remarkable improvement in the treatment of diverse cancer forms. Worldwide, good outcomes of the disease are estimated at 4% of the population [1,2]. However, the success of cancer treatment was achieved at a considerable cost [3,4,5]. This was connected with the properties of anthracyclines, discovered 60 years ago but still listed among the World Health Organization (WHO) recommended drugs for the treatment of childhood and adult cancer [4,6,7,8,9]

Acute cardiotoxicity of anthracyclines, especially the most commonly used doxorubicin (DOX), is observed during the first year of treatment. In pediatric patients, its incidence is low, owing to a cautious dosing regime, which limits drug cumulation. Furthermore, in young patients, pre-existing cardiovascular diseases (hypertension, hyperlipidemia), the recognized risk factors, are uncommon [10,11,12,13,14,15].

Unfortunately, in the long run, because of the cardiotoxicity of DOX the susceptibility of the treated patients to cardiac damage and the development of heart failure (HF) is dramatically higher (15fold) than in the untreated population [8,16,17]. This is observed both in children and adults. While it is common knowledge that one in eight women will develop breast cancer, we are only rarely aware that one in ten breast cancer patients whose chemotherapy regime includes DOX will develop cardiac damage and not infrequently DOX-induced HF [3,9,15]. Its incidence increases progressively, and the current standard therapeutic measures applied in DOX-induced HF prove less effective than in HF patients of other etiology [8,13,18,19,20].

Important progress has been made in effort to develop mitochondria-specific delivery of DOX (or similar anthracyclines) to tumor mitochondria, which would increase its accumulation in the tumor tissue and reduce exposure of the heart. Hence, it should result in a simultaneous increase in the efficiency and safety of DOX in cancer chemotherapy. Such attempt has been made by amphiphilic modification of DOX, which seems to be a very promising tool in the future [21]. However, despite intensive research and progress, to the best of our current knowledge, no mitochondria-targeting formulations of DOX have been approved for the clinical use. In addition, it is important to recognize that even after potential introducing of new modification of DOX, with expected minimal off-target toxicity, for many decades a large cohort of patients, particularly childhood cancer survivors, will be still endangered by the development of HF induced by the classic doxorubicin [7].

Hence, it is recognized that new treatment strategies for chemotherapy-induced HF are needed, however, the prerequisite for any success is the profound understanding of the pathophysiology of this HF form.

Without disregarding some obvious limitations, the small animal models of HF have proven to be invaluable tools that have greatly advanced our understanding of the pathophysiology of HF, and have helped to define new targets in the development of novel treatment strategies [22,23]. A number of mouse, rat, and rabbit models of cardiotoxicity induced by cancer treatment regimes (particularly DOX and trastuzumab, a monoclonal antibody applied in breast and stomach cancer) were proposed [24]. They were mostly employed to investigate the mechanisms and/or protective measures against chemotherapy-induced cardiotoxicity [24,25,26,27,28,29]. The effects of DOX on cardiac function and the development of HF were only of marginal interest [30,31,32,33]. It is worth mentioning that the value of other models of HF, such as ischemic injury (coronary artery ligation [34]) and non-ischemic HF (chronic volume overload induced by aorto-caval fistula (ACF) [35]) have been characterized in detail [22,23].

Considering an urgent need to further elucidate the pathophysiology of chemotherapy-induced HF and the insufficient in vivo characterization of a DOX-induced model of HF, the major aim of the present study was to evaluate cardiac morphological structure and the function parameters in rats with DOX-induced HF. Echocardiography and invasive pressure-volume analysis of the left ventricle (LV) were employed at the very early phase of HF development. In addition, the systemic and intrarenal activity status of neurohormonal systems were simultaneously evaluated.

Since hypertension and inappropriate activation of the renin-angiotensin-aldosterone system (RAAS) are considered as risk factors for the development of chemotherapy-induced cardiotoxicity, cardiomyopathy and ultimately chemotherapy-induced HF [11,15], we decided to explore the characteristics of the DOX-induced model of HF in Ren-2 transgenic rats (TGR), in which the endogenous activation of the RAAS and hypertension are combined [33,36,37], and to compare them with those of normotensive transgene-negative, Hannover Sprague-Dawley (HanSD) rats.

## 2. Results

### 2.1. Series 1: Echocardiographic Assessment of the Longitudinal Changes in Cardiac Function throughout DOX Administration

Figure 1 and Figure 2 summarize the results from evaluation of cardiac function and morphology by echocardiography. In HanSD rats the administration of DOX resulted in the decline in cardiac output, stroke volume, LV ejection fraction and in LV fraction shortening, without altering heart rates. All the above-mentioned changes were observed two weeks after the last injection (4 weeks values, Figure 1A–E). In the case of TGR, all the mentioned changes were also present, however, they were observed earlier and were also more pronounced, i.e., at the end of DOX administration (2 weeks values). Explicitly, two weeks after last injection (4 week values), DOX treatment caused in TGR significantly greater decreases than in HanSD rats in cardiac output (−34.6 ± 0.9 vs. −24.1 ± 0.7%), stroke volume (−25.9 ± 0.8 vs. −20.6 ± 0.4%), LV ejection fraction (−24.6 ± 0.6 vs. −15.9 ± 0.5%), and LV fractional shortening (−31.7 ± 2.4 vs. 13.1 ± 0.6%)—*p* < 0.05 in all cases.

As shown in Figure 2A there was no significant difference in the LV diastolic diameter between TGR and HanSD rats under basal conditions, and administration of DOX did not change it in any of experimental groups or at any time point. The hypertensive TGR under basal conditions revealed, under basal conditions, markedly higher LV anterior and posterior wall thickness as compared with normotensive HanSD rats (both in diastole and systole) (Figure 2B–E). DOX treatment resulted in reduction of LV wall thickness in both TGR and HanSD rats as compared with the untreated counterparts (Figure 2B–E), as observed two weeks after termination of the treatment (4 week values). This effect was more distinct for LV posterior wall thickness (both in diastole and systole), than in the anterior wall (significant only in systole). In addition, as shown in Figure 2F, two weeks after the last injection (4 week values) DOX was seen to cause a significant decrease in the relative wall thickness in TGR. Moreover, DOX treatment two weeks after the last injection (4 week values) resulted in TGR a more pronounced reduction of wall thicknesses than in HanSD rats, explicitly in LV anterior wall thickness in systole (−20.6 ± 0.5 vs. −14.1 ± 0.4%), LV posterior wall thickness in diastole (−23.8 ± 0.5 vs. −11.1 ± 0.3%), and LV posterior wall thickness in systole (−28.1 ± 0.6 vs. −11.3 ± 0.3%)—*p* < 0.05 in all cases.

Figure 3 and Figure 4 summarize the effect of DOX on body weight, organ weights and plasma albumin levels two weeks after the last DOX dose (i.e., 4 week values; at the end of the experimental protocol), both in HanSD rats and TGR. The data are pooled from all three series of experiments.

As shown in Figure 3A,B, DOX did not cause significant decreases in body weight and kidney weight in HanSD rats, but did so in TGR. Moreover, DOX caused similar decreases in plasma albumin levels in HanSD rats and TGR (Figure 3C).

Figure 4A,B show that untreated TGR exhibited markedly higher whole heart and LV weights as compared with untreated HanSD rats, but there were no significant differences in the RV weights between them. DOX treatment elicited significant and proportional decreases in the whole heart, LV and RV weights as seen from equal ratios of the RV to LV weights with and without DOX administration (Figure 4A–D). As shown in Figure 4E,F, DOX did not alter lung weight (wet or dry) in HanSD rats but significantly decreased both parameters in TGR.

### 2.2. Series 2: Assessment of Cardiac Function with Invasive Hemodynamic Pressure-Volume Method

Figure 5 and Figure 6 summarize the data from evaluation of cardiac function by the invasive hemodynamics method. The administration of DOX to HanSD rats caused only significant decrease in LV ejection fraction in comparison to untreated counterparts (Figure 6C). In contrast, administration of DOX to TGR resulted not only in reduced LV ejection fraction (Figure 6C), but also in the significant impairment in maximum rates of pressure rise (+dP/dt)_max_ and maximum rates of pressure fall (−dP/dt)_max_ (Figure 5D,E). In addition, the administration of DOX elicited significant decreases in end-systolic pressure volume relationship (ESPVR) and in preload recruitable stroke work (PRSW) in TGR as compared with untreated counterparts (Figure 6A,D), which was accompanied by significant increase in ventricular-arterial coupling (Figure 6F).

Figure 7 shows representative steady-state loops from the pressure-volume analysis in untreated and DOX-treated HanSD rats and TGR. Inspection of the loops further supports the notion that TGR after DOX administration exhibit substantial impairment of systolic function when compared with the untreated counterparts.

Figure 8 summarizes the results from echocardiographic evaluation of the RV performed during the invasive hemodynamics analysis. As shown, there were no significant differences in the RV diastolic diameter and RV ejection fraction between untreated TGR and HanSD rats. The administration of DOX did not alter the RV diastolic diameter, either in TGR or HanSD rats. However, in both strains, it caused a significant and proportional decrease in RV ejection fraction (Figure 8B).

### 2.3. Series 3: Assessment of the Effects of DOX Administration on Plasma and Kidney Concentrations of Angiotensin II (ANG II), Angiotensin 1-7 (ANG 1-7) and Norepinephrine (NE)

Figure 9A shows that plasma NE levels in untreated HanSD rats and TGR were almost identical and that administration of DOX elicited significant increases in HanSD rats, but did not alter them in TGR. In contrast, renal NE concentrations were significantly higher in untreated HanSD rats when compared with untreated TGR, but the administration of DOX did not change renal NE concentrations, similarly in HanSD rats or in TGR (Figure 9E). Figure 9B,F show that plasma and the whole kidney ANG II levels were about twofold higher in untreated TGR than in untreated HanSD rats. Administration of DOX resulted in marked increases in plasma and kidney ANG II levels in both HanSD rats and TGR, but still the difference between the two strains were maintained (significantly higher concentrations in TGR as compared with HanSD rats after DOX treatment).

Figure 9C,G show that there were no significant differences in plasma and kidney ANG 1-7 levels between untreated HanSD rats and TGR. DOX administration elicited significant increases in plasma and kidney ANG 1-7 concentrations in HanSD rats as well as in TGR, but they were more prominent in TGR.

Figure 9D,H help to assess the systemic and intrarenal balance between vasodilator and vasoconstrictor axes of the RAAS expressed as the ratio of ANG 1-7 to ANG II. In previous studies, including our own [38], this ratio has been validated as a reliable marker of the activity of the vasodilator axis of RAAS under conditions of vasoconstrictor axis hyperactivity.

Substantially lower values of this index were observed in plasma and kidneys of untreated TGR versus untreated HanSD rats (about twofold and threefold lower, respectively). In HanSD rats, the administration of DOX did not change the ratio of ANG 1-7 to ANG II, similarly in systemic and intrarenal compartments. In contrast, DOX administration resulted in significant increases in this index in TGR, and in the kidney its values reached the level observed in the untreated HanSD rats.

## 3. Discussion

The first important finding of the present study is that, two weeks after the end of DOX treatment, normotensive HanSD rats showed an impairment of cardiac function. Moreover, all the changes then observed closely resembled “chemotherapy-induced HF with reduced ejection fraction (HFrEF)” described in humans [4,5,6,7,8,12,13,17,18,19,39]. In HanSD rats, DOX caused bilateral cardiac atrophy and impairment of the liver synthetic function (see reduced plasma albumin levels) without affecting the whole body or other organs’ weight. Remarkably, the finding of marked cardiac atrophy agrees well with the recent report by Jordan et al. [40] who clearly demonstrated that the recipients of anthracycline-based chemotherapy showed a 5% decline in myocardial mass as early as six months post treatment, even under conditions of increased afterload The study by Jordan et al. [40] was a milestone in the field because, up to this point, most of the surveillance strategies for the detection of early signs of chemotherapy-induced HF with HFrEF were focused on a serial assessment of LV ejection fraction to identify LV systolic dysfunction but did not include LV mass evaluation [41,42].

Of particular interest were also the data on systemic and intrarenal RAAS and SNS activities. In HanSD rats, DOX activated the systemic (but not intrarenal) SNS. Also enhanced were the systemic and intrarenal activity of the vasoconstrictor and vasodilator axes of the RAAS, however, the balance of the two axes was maintained. Given the recent finding that an elevated ANG 1-7/ANG II ratio predicts a beneficial outcome of HF [43], it is reassuring that at least the ratio was not lowered. Importantly, our biochemical findings indicate that the compensatory activation of neurohormonal systems was at the initial stage. It is now agreed that such activation is initially beneficial. However, if prominent and long-lasting, it might be extremely deleterious and substantially contribute to the progression of HF, which makes it a life-threatening disorder [44,45,46,47]. Taken together, our present functional and biochemical data indicate that, two weeks after the termination of DOX treatment, HanSD rats are at the very early phase of chemotherapy-induced HFrEF, with the dominant symptom of bilateral cardiac atrophy.

The second important set of findings is that, compared to HanSD rats, in the TGR model already at the end of the treatment DOX significantly decreased cardiac output with substantial impairment of systolic function, and these alterations progressed throughout the two weeks after cessation of the treatment. Moreover, in addition to bilateral cardiac atrophy and impairment of the liver synthetic function as observed in HanSD rats, DOX significantly decreased the whole BW and the weight of the kidney and lungs. Furthermore, besides LV atrophy DOX administration in TGR caused LV dilatation as indicated by a significant decrease in LV relative wall thickness. We noticed also that TGR subjected to DOX treatment did not show systemic and intrarenal activation of SNS, whereas there was a marked activation of both axes of the RAAS. Interestingly, based on the significant increase in plasma and kidney ANG 1-7/ANG II ratio, activation of the vasodilator axis of RAAS appeared more pronounced. As reported recently and already mentioned above [42], the increased ANG 1-7/ANG II ratio might be considered as the beneficial factor slowing down the progression of HF.

Our results indicate that, similarly to HanSD rats, TGR developed chemotherapy-induced HFrEF after DOX treatment. However, the substantial impairment of LV contractility, as apparent from decreased (+dP/dt)_max_, ESPVR, and PRSW, was more pronounced than in HanSD rats. Remarkably, despite such an apparent impairment of LV contractility in TGR, when LV ejection fraction was assessed by the invasive hemodynamic pressure-volume method, the reduction was not more pronounced in TGR than in HanSD rats, as observed when cardiac function was evaluated by echocardiography. We cannot provide a fully satisfactory explanation for this discrepancy, but we assume that it is elicited by methodological factors, in particular by different anesthesia’s employed for these two methods. Nevertheless, despite this discrepancy, overall, our findings are in agreement with our original hypothesis that hypertension and augmented activation of the RAAS might accelerate the onset of DOX-induced HF, and support the previously expressed notion that hypertension and over-activation of the RAAS constitute major risk factors for the development of chemotherapy-induced cardiomyopathy and HF [11,12,13,14,15,17,19,39].

In addition, our present findings extend those of Sharkey et al. [48] who found that the delayed toxic effects of DOX treatment were exacerbated in genetically hypertensive spontaneously hypertensive rats (SHR) and in rats genetically predisposed to develop hypertension combined with cardiomyopathy. In general, our present findings strongly support the widespread view that a crosstalk between the common risk factors can predispose to both cancer and cardiovascular diseases, particularly to heart failure [49,50,51].

Taking together the assessment of cardiac function, organ morphology and biochemical data from our current study and our previous results from the ACF-induced HFrEF model in TGR [52,53,54], we propose that after DOX treatment TGR exhibit signs of HFrEF in the phase of compensation resembling those in HanSD rats but with dominant bilateral cardiac atrophy combined with impairment of systolic functions.

On the whole, while admitting the limitations of small animal models of HF, they remain an irreplaceable tool for improving our understanding of various aspects of pathophysiology of HF and help us develop novel treatment strategies. Our current results indicate that the model of DOX-induced HFrEF strongly resembles clinical situation in patients with chemotherapy-induced HF. As defined more than 20 years ago, an optimal animal model of human cardiovascular disease should (i) mimic the human disease, (ii) allow studies in chronic, stable disease, (iii) produce symptoms which are predictable and controllable, (iv) satisfy economical, technical and animal welfare considerations, and (v) allow relevant measurement of cardiac, hemodynamic and biochemical parameters [55]. Based on our current results and recent reports we believe that all these prerequisites are fulfilled in HanSD rats, as well as in TGR two weeks after the cessation of DOX administration. Nevertheless, it should be emphasized that by this time TGR had displayed signs of compensated chemotherapy-induced HFrEF. However, a closer scrutiny of some data, especially of the analysis of the invasive hemodynamic pressure-volume studies, suggests that the rats were at the beginning of the transition stage from the compensated to the decompensated phase of HF. It is important to acknowledge that, in order to undoubtedly define the stage of transition from the compensated to the decompensated phase of DOX-induced HF in TGR (as well as in HanSD rats), long term studies evaluating HF-related morbidity and mortality are necessary. We performed similar studies that characterized the course of HF development in an ACF model in TGR and HanSD rats [51,56]. However, such studies are very challenging. We characterized the course of ACF-induced HF in normotensive rats and we found that first deaths occur around the 20th week and the median survival was around 43 weeks after ACF induction [56] and even if the onset of the phase of decompensation in TGR after ACF creation is markedly accelerated still the median survival is about five weeks [52,54,57,58]. It is plausible to assume that, in DOX-induced HFrEF, more long term studies will be also required and it is apparent that future studies are needed to precisely define the phases of compensated and decompensated chemotherapy-induced HFrEF in TGR as well as in HanSD rats. Nevertheless, despite this limitation, on the whole, the present results strongly support the view that DOX-induced HFrEF, particularly in TGR, is an optimal model to study the pathophysiological aspects of chemotherapy-induced HFrEF. Therefore, the model should be applied for evaluation of the urgently needed novel therapeutic strategies to combat this condition.

## 4. Methods

### 4.1. Experimental Animals and HF Induction

All animals used in the present study were bred at the animal house of the Center of Experimental Medicine, Institute for Clinical and Experimental Medicine (IKEM, Prague, Czech Republic), which is accredited by the Czech Association for Accreditation of Laboratory Animal Care. Heterozygous TGR were generated by breeding male homozygous TGR with female homozygous HanSD rats and age-matched HanSD rats served as controls. The animals were kept on a 12 h/12 h light/dark cycle. Throughout the experiments rats were fed a normal salt, normal protein diet (0.45% NaCl, 19–21% protein) manufactured by SEMED (Prague, Czech Republic) and had free access to tap water. Male TGR and HanSD rats, at the initial age of 8–9 weeks, derived from several litters, were randomly assigned to the experimental groups.

A simple and well-established procedure for DOX-induced cardiomyopathy followed by HF was employed, involving six intraperitoneal (i.p.) injections of DOX (Doxorubicin Ebewe 2mg/ml, Ebewe Pharma, Unterach, Austria), (2.5 mg/kg of body weight) over two weeks, resulting in cumulative dose of 15 mg/kg of body weight. This method has been frequently used for more than 30 years [59] and besides its simplicity, it reflects well the human clinical circumstances; the cumulated dose in our rats corresponds to 550–600 mg/m^2^ body surface applied in patients. The incidence of DOX-induced cardiomyopathy with this dose is usually 18% and even higher in hypertensive patients [8,13,14,15,16,17,18,60]. Control animals received vehicle solution in the same volume (saline solution with lactose at the same concentration as used for dilution of DOX). It is evident that our experimental approach combines the advantage of a very simple experimental procedure with high clinical relevance.

### 4.2. Experimental Design

#### 4.2.1. Series 1: Echocardiographic Assessment of the Longitudinal Changes in Cardiac Function throughout DOX Administration

The aim of this series was to assess the changes in cardiac morphology during DOX treatment. Echocardiographic examination was performed three days before the first DOX administration (basal values), at the end of 2 weeks’ DOX treatment and then after additional two weeks (4 weeks values). It has been suggested that this time is sufficient to fully develop the signs of DOX-induced cardiomyopathy [23,24,30,31,59]. Echocardiographic assessment is described in detail in our recent study [61]. Briefly, the animals were anesthetized with 4% isoflurane combined with 3 L/min oxygen, the ventral thorax was shaved. During the image acquisition, the rats were maintained under isoflurane anesthesia (2–2.3%, at oxygen flow of 1 L/min, if necessary the dosage was slightly adjusted, depending on the animal’s weight, its reaction and breathing), and fixed in the supine position. For standard measurements of cardiac parameters, B-MODE and M-Mode images were recorded in parasternal long axis and parasternal short axis view at the papillary muscle level. Morphological parameters of the LV, including dimension of LV inner diameter, anterior and posterior walls were measured in M-mode from long and short axis sections as previously described [61]. All ultrasound studies were done by Vevo^®^ 2100 Imaging System with the MS250S transducer (13–24 MHz), (FUJIFILM VisualSonics, Inc., Toronto, ON, Canada). For each parameter, the mean of 3 optimally obtained measurements was used. The following experimental groups of animals were examined:HanSD rats + vehicle (*n* = 8)TGR + vehicle (*n* = 7)HanSD rats + DOX (*n* = 8)TGR + DOX (*n* = 8)

In addition, at the end of each experiment whole heart, LV (assessed as LV + septum), right ventricle (RV), liver, kidney and lung weights were measured.

#### 4.2.2. Series 2: Assessment of Cardiac Function with Invasive Hemodynamic Pressure-Volume Method

The aim of this series was to evaluate the cardiac function by LV pressure-volume analysis two weeks after the last DOX administration. Based on recent reports [32,33] and the results from Series 1 of the present study, it is known that at this stage the animals show signs of cardiac dysfunction. The goal was to characterize its degree and to confirm that the rats exhibit signs of compensated HF. DOX- and vehicle-treated TGR and HanSD rats were prepared as ascribed above. Two weeks after the last injection, animals were anesthetized with ketamine/midazolam combination (50 and 5 mg/kg of body weight, respectively; i.p.) and echocardiography and subsequently invasive hemodynamic evaluation was performed as described in detail in our previous studies [52,62].

Rats were divided into the following experimental groups:HanSD rats + vehicle (*n* = 12)TGR + vehicle (*n* = 12)HanSD rats + DOX (*n* = 14)TGR + DOX (*n* = 15)

At the end organ weights were assessed as in Series 1.

#### 4.2.3. Series 3: Assessment of the Effects of DOX Administration on Plasma and Kidney ANG II, ANG 1-7 and NE Concentrations

The aim of this series was to assess the degree of activation of two axes of RAAS: the vasoconstrictor axis, represented by ANG II concentration and the vasodilatory axis, represented by ANG 1-7 concentration, together with the determination of the sympathetic nervous system (SNS) activation, represented by NE concentration. It is well known that ANG II concentration under anaesthesia is higher than in conscious rats and there are also marked differences in the renin secretion in response to anaesthesia and surgery per se and in the activation of the RAAS between TGR and HanSD rats. Therefore ANG II, ANG 1-7, and catecholamine concentrations were measured in samples from decapitated animals by methods described in detail in our recent studies [38,53,54,63].

Animals were exposed to the same experimental protocol as in Series 1 and 2, and were decapitated two weeks after the last injection of DOX (the same time schedule as in Series 2). The same experimental groups were evaluated as in Series 1 and 2 (*n* = 8 in each group)

### 4.3. Statement of Ethics

The study followed the guidelines and practices established by the Animal Care and Use Committee of the IKEM, which accord with the national law and with American Physiological Society guiding principles for the care and use of vertebrate animals in research and training, and were approved by the Animal Care and Use Committee of the IKEM and, consequently, by the Ministry of Health of the Czech Republic (project decision 36388/2019-4/).

### 4.4. Statistical Analysis

Statistical analysis of the data was performed using Graph-Pad Prism software (Graph Pad Software, San Diego, CA, USA). Statistical comparison of other results was made by Student’s *t*-test, Wilcoxon´s signed-rank test for unpaired data or one-way ANOVA when appropriate. Values are expressed as mean ± SEM. The values of *p* below 0.05 were considered statistically significant.

## Figures and Tables

**Figure 1 ijms-21-09337-f001:**
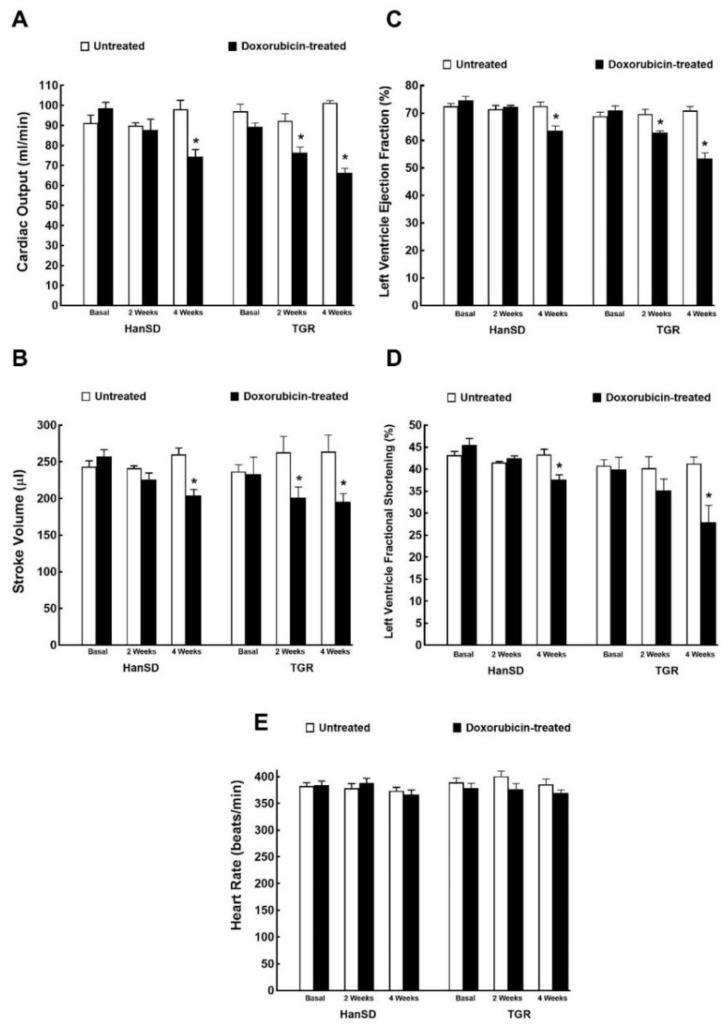
Assessment of the cardiac functional parameters by echocardiography before administration of doxorubicin (basal values), at the end of doxorubicin administration (2 weeks values) and two weeks after termination of doxorubicin treatment (4 weeks values) in normotensive, transgene-negative Hannover Sprague-Dawley (HanSD) and hypertensive, Ren-2 transgenic (TGR) rats. Cardiac output (**A**), stroke volume (**B**), left ventricle ejection fraction (**C**), left ventricle fraction shortening (**D**), heart rate (**E**). * *p* < 0.05 compared with untreated animals of the same strain at the same time point. The values are means ± SEM.

**Figure 2 ijms-21-09337-f002:**
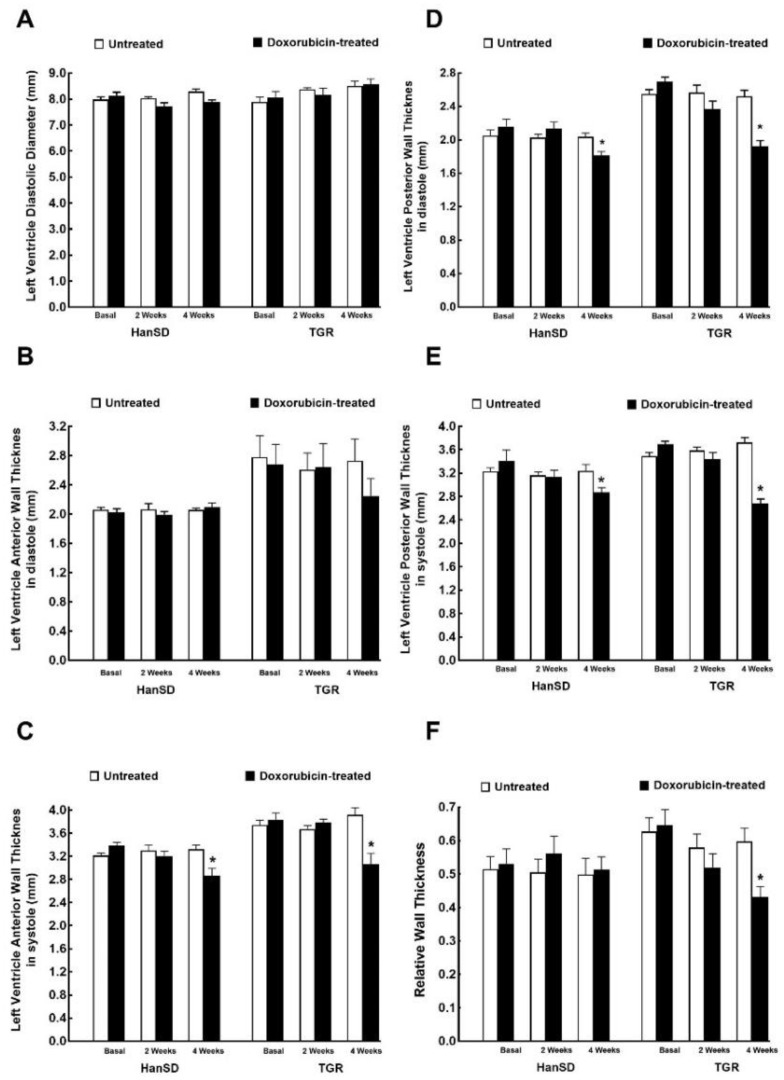
Assessment of the cardiac morphological parameters by echocardiography before administration of doxorubicin (basal values), at the end of doxorubicin administration (2 weeks values) and two weeks after termination of doxorubicin treatment (4 weeks values) in normotensive, transgene-negative Hannover Sprague-Dawley (HanSD) and hypertensive, Ren-2 transgenic (TGR) rats. Left ventricle diastolic diameter (**A**), left ventricle anterior wall thickness in diastole (**B**), left ventricle anterior wall thickness in systole (**C**), left ventricle posterior wall thickness in diastole (**D**), left ventricle anterior wall thickness in (**E**), relative wall thickness (**F**). * *p* < 0.05 compared with untreated animals of the same strain at the same time point. The values are means ± SEM.

**Figure 3 ijms-21-09337-f003:**
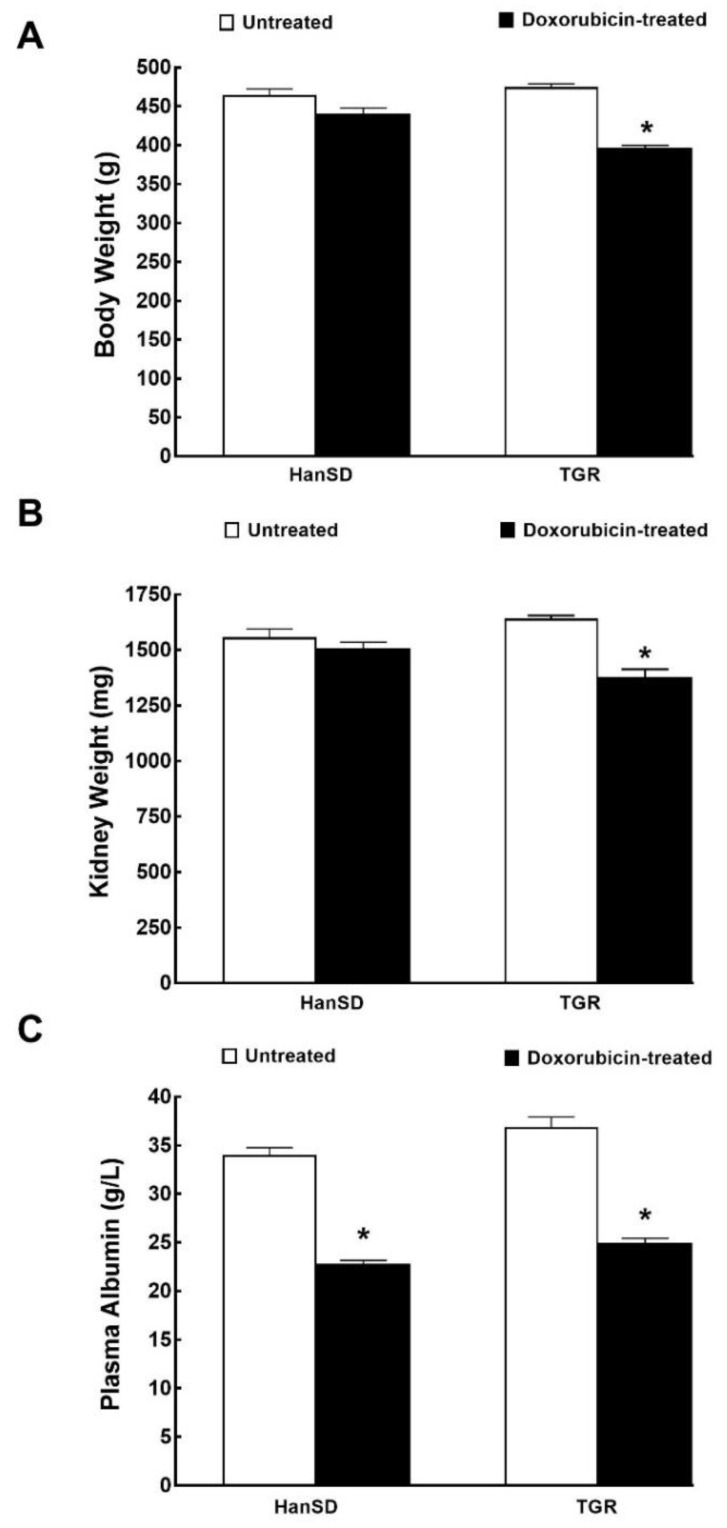
Weights and biochemical parameter. Body weight (**A**), kidney weight (**B**) and plasma albumin levels (**C**) two weeks after the last doxorubicin injection (at the end of the experimental protocol) in normotensive, transgene-negative Hannover Sprague-Dawley (HanSD) and hypertensive, Ren-2 transgenic (TGR) rats. * *p* < 0.05 compared with untreated animals of the same strain at the same time point. The values are means ± SEM and are pooled from the three series of experiments.

**Figure 4 ijms-21-09337-f004:**
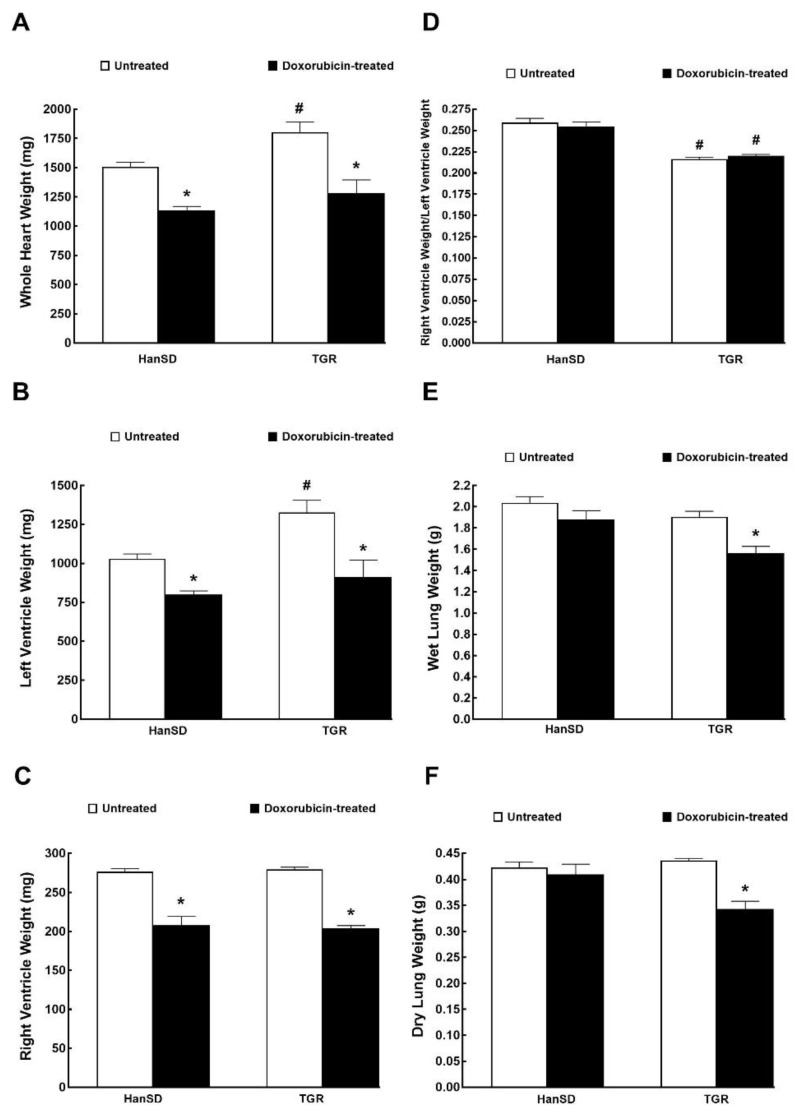
Whole heart weight (**A**), weights of individual components of the heart (**B**,**C**), their ratio (**D**) and wet and dry lung weights (**E**,**F**) two weeks after the last doxorubicin injection (i.e., at the end of the experimental protocol) in normotensive, transgene-negative Hannover Sprague-Dawley (HanSD) and hypertensive, Ren-2 transgenic (TGR) rats. * *p* < 0.05 compared with untreated animals of the same strain. ^#^
*p* <0.05 versus HanSD rats within the same protocol. The values are means ± SEM and are pooled from the three series of experiments.

**Figure 5 ijms-21-09337-f005:**
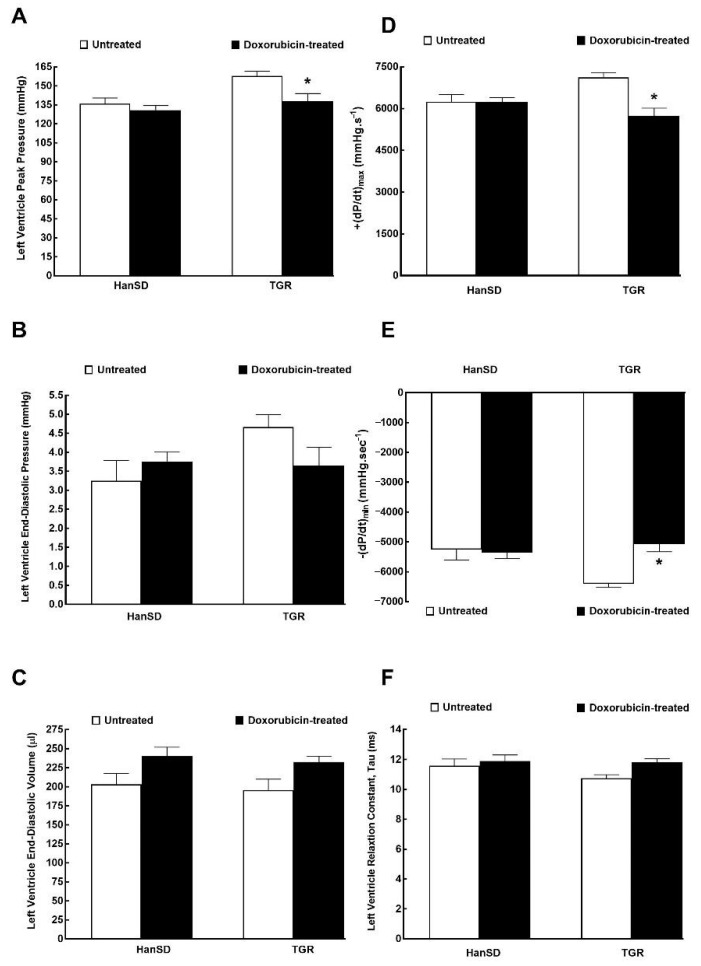
The first data part of the left ventricular cardiac function assessment by invasive hemodynamic analysis performed two weeks after the last doxorubicin injection (i.e., at the end of the experimental protocol) in normotensive, transgene-negative Hannover Sprague-Dawley (HanSD) and hypertensive, Ren-2 transgenic (TGR) rats. Left ventricle peak pressure (**A**), left ventricle end-diastolic pressure (**B**), left ventricle end-diastolic volume (**C**), maximum rates of pressure rise (+dP/dt)_max_ (**D**), maximum rates of pressure fall (−dP/dt)_max_ (**E**), relaxation constant tau (**F**). * *p* < 0.05 compared with untreated animals of the same strain. The values are means ± SEM.

**Figure 6 ijms-21-09337-f006:**
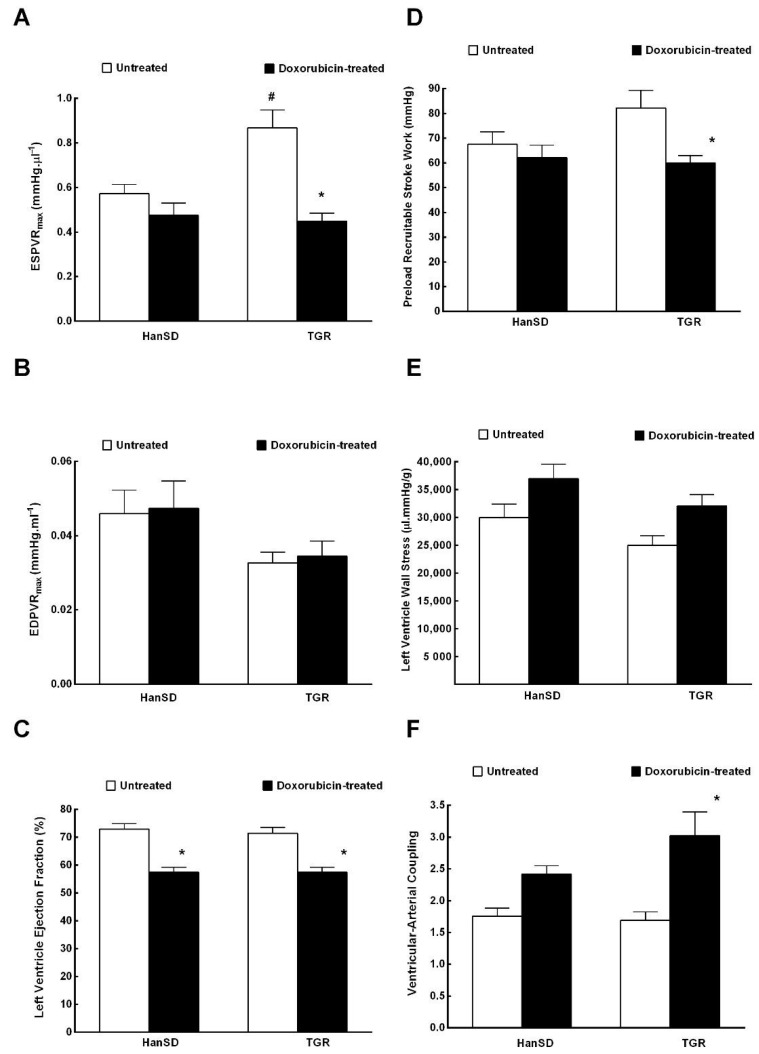
The second part of the data for the left ventricular cardiac function assessment by invasive hemodynamic analysis two weeks after the last doxorubicin injection (i.e., at the end of the experimental protocol) in normotensive, transgene-negative Hannover Sprague-Dawley (HanSD) and hypertensive, Ren-2 transgenic (TGR) rats. End-systolic pressure volume relationship (ESPVR) (**A**), end-diastolic pressure volume relationship (EDPVR) (**B**), left ventricle ejection fraction (**C**), preload recruitable stroke work (**D**), left ventricle wall stress (**E**), ventricular-arterial coupling (**F**). * *p* <0.05 compared with untreated animals of the same strain. The values are means ± SEM. ^#^
*p* < 0.05 compared with untreated HanSD rats.

**Figure 7 ijms-21-09337-f007:**
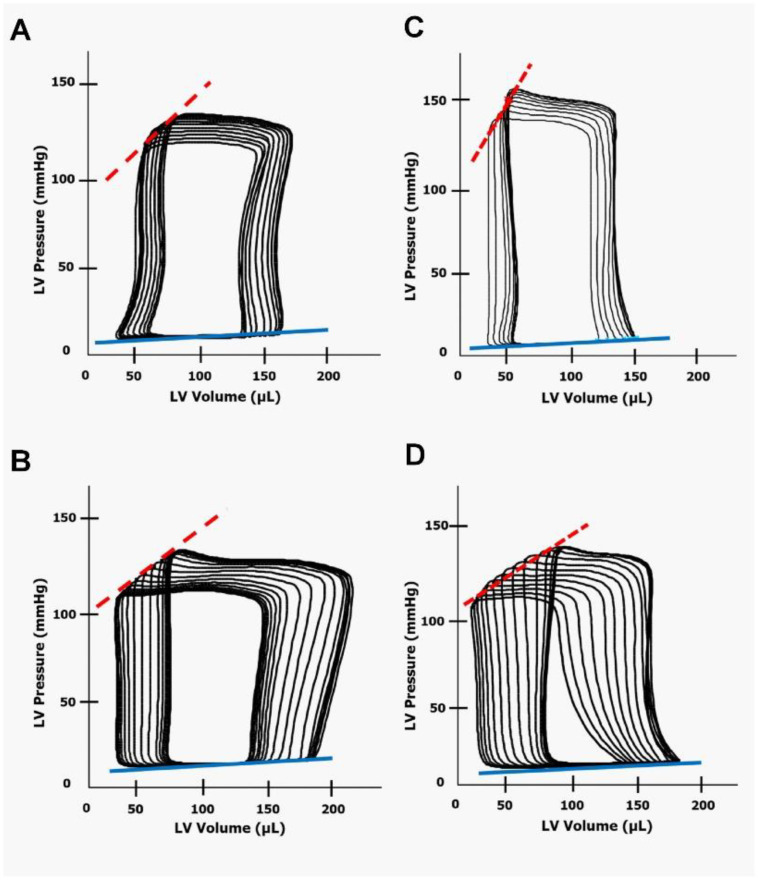
Representative steady-state loops from the pressure-volume analysis performed in (**A**) untreated normotensive, transgene-negative Hannover Sprague-Dawley (HanSD) rats, (**B**) HanSD rats two weeks after the last doxorubicin injection (i.e., at the end of the experimental protocol), (**C**) untreated hypertensive, Ren-2 transgenic (TGR) rats and (**D**) TGR after doxorubicin administration. ESPVR, end-systolic pressure volume relationship (dotted red line). EDPVR, end-diastolic pressure volume relationship (blue line).

**Figure 8 ijms-21-09337-f008:**
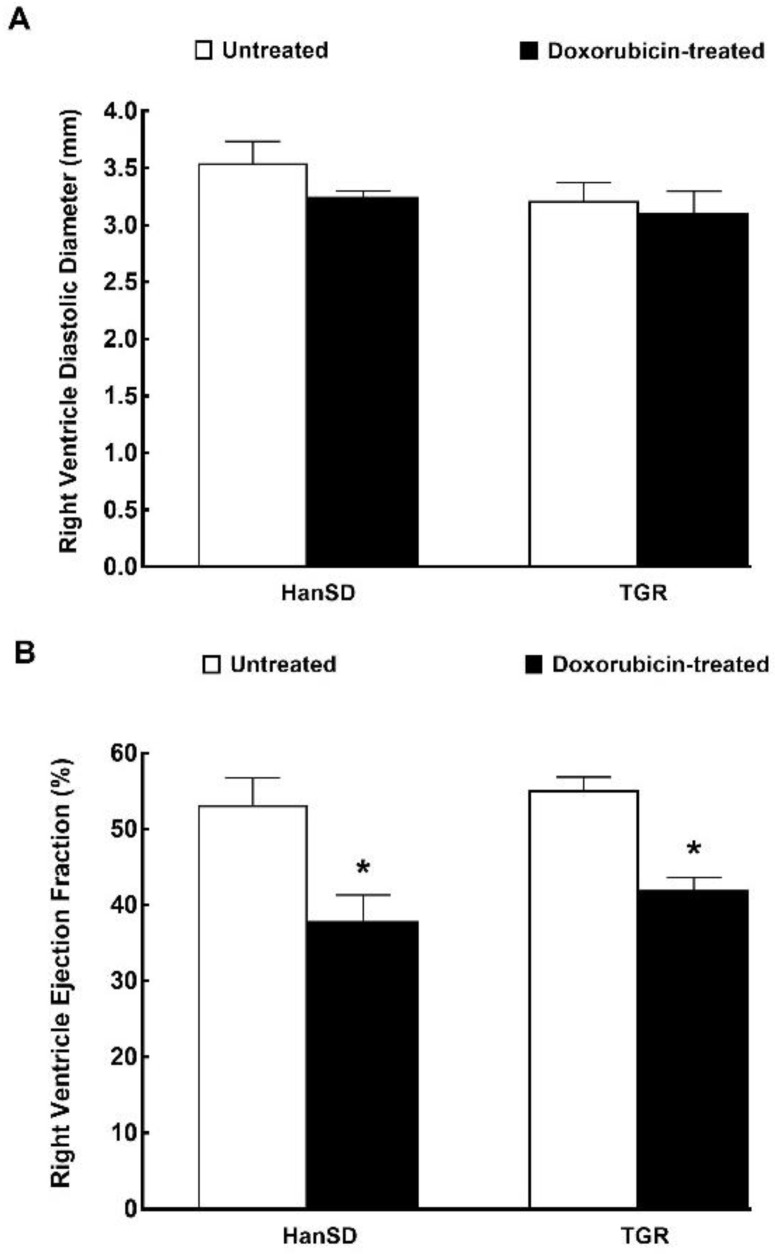
Right ventricle function. Assessment of the right ventricle diastolic diameter (**A**) and right ventricle ejection fraction (**B**) by echocardiography after the last doxorubicin injection (i.e., at the end of the experimental protocol) in normotensive, transgene-negative Hannover Sprague-Dawley (HanSD) and hypertensive, Ren-2 transgenic (TGR) rats. * *p* < 0.05 compared with untreated animals of the same strain. The values are means ± SEM.

**Figure 9 ijms-21-09337-f009:**
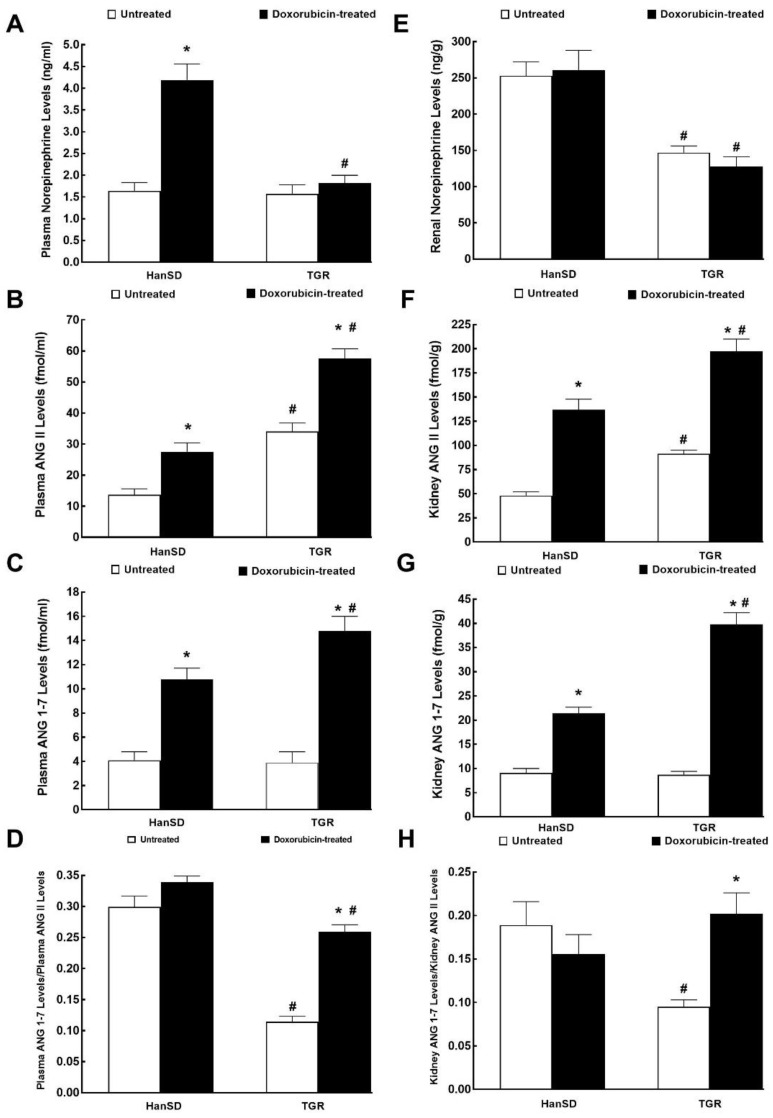
Plasma and kidney biochemical parameters. Plasma and kidney norepinephrine (**A**,**E**), angiotensin II (ANG II) (**B**,**F**), angiotensin 1-7 (ANG 1-7) (**C**,**G**) levels and the ratio of ANG 1-7 to ANG II (**D**,**H**) after the last doxorubicin injection (i.e., at the end of experimental protocol) in normotensive, transgene-negative Hannover Sprague-Dawley (HanSD) and hypertensive, Ren-2 transgenic (TGR) rats. * *p* < 0.05 compared with untreated animals of the same strain. ^#^
*p* < 0.05 versus HanSD rats exposed to the same protocol. The values are means ± SEM.

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
