# Peer review of "Deleterious Effects of Hyperactivity of the Renin-Angiotensin System and Hypertension on the Course of Chemotherapy-Induced Heart Failure after Doxorubicin Administration: A Study in Ren-2 Transgenic Rat"

_ijms, 2020, doi:10.3390/ijms21249337_

Round 1
Reviewer 1 Report
The research article deals with the evaluation of the influence of doxorubicin-induced heart failure in a Ren-2 transgenic rat model (TGR), characterized by hypertension and hyperactivity of the renin-angiotensin-aldosterone system. The observations in TGR rats were compared with the use of control, normotensive Hannover Sprague-Dawley (HanSD) rats. The results showed that in HanSD rats DOX induced early phase heart failure with reduced ejection fraction(HFrEF) two weeks post-drug dosage, with the dominant symptom of bilateral cardiac atrophy. In TGR rats, decreased cardiac output with impairment of systolic function was observed earlier, at the termination of drug administration. The impairment of LV contractility was more pronounced in TGR rats. Based on the results the authors concluded that hypertension and increased RAAS can accelerate the onset of DOX-induced heart failure and that the TGR rat model is suitable for implementing in the study of the pathophysiological aspects of this drug-induced condition.
Overall, the manuscript is well-written and clearly presented. I have the following questions for the authors regarding their article:
The authors mention that cardiac function changes in TGR rats were more pronounced after DOX administration in comparison to HanSD rats (Fig 1). Was the significance of the decrease in cardiac functions compared between these two groups?
The authors suggest, based on the obtained results, that two weeks after the cessation of DOX administration in TGR rats, the animals were at the beginning of the transition stage from the compensated to the decompensated phase of heart failure. Did the authors consider prolonging the duration of the observation of the post-treatment effects of DOX on TGR rats in order to evaluate the long-term influence and delayed toxic effects of DOX on this drug-induced HF model.
The manuscript is well-written, however some minor errors are present: A few are listed below:
Line 22: The aim of the present study
Line 26: in a cumulative dose of
Line 81: explore the characteristics of the
Line 244: in the TGR model
Line 245: DOX significantly
Line 276: remain an irreplaceable
Author Response
We appreciate the reviewers´ comments on our study and are grateful for their encouraging and constructive suggestions.
Responses to reviewer #1:
Reviewers Comment 1: The authors mention that cardiac function changes in TGR rats were more pronounced after DOX administration in comparison to HanSD rats (Fig 1). Was the significance of the decrease in cardiac functions compared between these two groups?
Authors Response ad point 1:
Yes, we compared the effects of DOX administration on cardiac function (Figure 1) and cardiac morphology (Figure 2) between TGR and HanSD rats. After some consideration, we decided to incorporate these results into the Results section of the revised manuscript. We believe that they support our notion that changes in TGR were more pronounced as compared with HanSD rats (page 7, line 10 to 14 and also page 7, line 25 to 29, shown in red font).
Additionally, we also decided to discuss in more detail the issue of certain discrepancies in the effects of DOX treatment on LV ejection fraction, when measured with two different methods, i.e. echocardiography and invasive hemodynamic pressure-volume method, which was probably noticed by the Reviewer but kindly not mentioned. Even though it is only a slight difference we would like to acknowledge this issue. Despite the apparent impairment of LV contractility in TGR, when LV ejection fraction was assessed by an invasive hemodynamic pressure-volume method the reduction was not more pronounced in TGR than in HanSD rats as observed when cardiac function was evaluated by echocardiography. We cannot provide a fully satisfactory explanation for this discrepancy, but we assume that it is elicited by methodological factors, i.a. by different anaesthesia’s employed for these two methods. We discuss this issue now in the Discussion section of the revised manuscript (page 10, line 11 to 16, shown in red font).
Reviewers Comment 2: The authors suggest, based on the obtained results, that two weeks after the cessation of DOX administration in TGR rats, the animals were at the beginning of the transition stage from the compensated to the decompensated phase of heart failure. Did the authors consider prolonging the duration of the observation of the post-treatment effects of DOX on TGR rats in order to evaluate the long-term influence and delayed toxic effects of DOX on this drug-induced HF model.
Authors Response ad point 2:
We recognize that the reviewer raised a very important issue because, in order to undoubtedly define the stage of compensated and decompensated phase of HF and particularly to define the transition stage, long term experiments focused on the morbidity and mortality are needed, as we performed when we characterized the rats’ model with volume overload due to aorto-caval fistula. We discuss this limitation of our study in the Discussion section of the revised manuscript (page 10, line 15 to 27, shown in red font).
Reviewers Comment 3: The manuscript is well-written, however, some minor errors are present: A few are listed below:
Line 22: The aim of the present study
Line 26: in a cumulative dose of
Line 81: explore the characteristics of the
Line 244: in the TGR model
Line 245: DOX significantly
Line 276: remain an irreplaceable
Authors Response ad point 3:
We are sorry for the grammatical and typing errors and we are thankful for pointing them out by the Reviewer. We carefully went throughout the manuscript in order to correct them.
Reviewer 2 Report
Kala et al studied the side effects of doxorubicin in cardiac failure in TGR and HanSD rats. The results showed clear significant difference in these two rat models in terms of DOX-induced morbidity. It could be an interesting addition to the journal for chemotherapy-induced cardiac study. However, there are some minor issues that may need the authors attention.
- The abstract is redundant regarding animal injection schedule, while insufficient key points are highlighted since the authors have generated much data.
- The figures layout can be further improved for quality control.
- Recently, researchers have showed a new doxorubicin conjugate that significantly reduces side effects on heart. (doi.org/10.1021/acsami.8b17399) The authors are encouraged to have discussion on this new doxorubicin conjugate and defend the necessity to study the doxorubicin-related side effects on heart.
Author Response
We appreciate the reviewers´ comments on our study and are grateful for their encouraging and constructive suggestions.
Responses to Reviewer #2:
Reviewers Comment 1: The abstract is redundant regarding animal injection schedule, while insufficient key points are highlighted since the authors have generated much data.
Authors Response ad point 1:
We recognize that a large number of important data is not presented in the abstract, however, it is more due to a space limitation. We agree that the information regarding “animal injection schedule” was really redundant and the abstract was modified as suggested.
Reviewers Comment 2: The figures layout can be further improved for quality control.
Authors Response ad point 2:
We agree that the quality of figures in the PDF was not good enough, however, it was automatically generated by the editorial system. We revised all figures in an effort to improve their quality and it seems that JPG format, which we used to prepare original figures from Graph Pad software, provides sufficient quality. We believe that eventually in the final version of the manuscript figures will be in the higher quality than were in PDF format for the review process.
Reviewers Comment 3: Recently, researchers have shown a new doxorubicin conjugate that significantly reduces side effects on the heart. (doi.org/10.1021/acsami.8b17399) The authors are encouraged to have a discussion on this new doxorubicin conjugate and defend the necessity to study the doxorubicin-related side effects on the heart.
Authors Response ad point 3:
This is a very interesting issue and we are grateful for pointing it out. The amphiphilic doxorubicin (amph-DOX) unquestionably represents a very promising new conjugate. However, in the above-mentioned study by Jingchao Xi et al. on amph-DOX, the authors admit that despite the favourable cardiosafety profile in the preclinical model, the long-term cardiotoxicity cannot be determined in the employed model.
Additionally to our knowledge, despite intensive research and progress, it has not been approved for the clinical use yet. Therefore, at least for the next decade, a large cohort of childhood cancer survivors (it is estimated that by the year 2020 it will be approximately 500 000 of them; reference number 7 in the revised manuscript) will be endangered by the HF development. In our opinion, this is still a very important rationale to study the pathophysiology of “chemotherapy-induced HF”. We provide this rationale in the Introduction section of the revised manuscript (from page 3, line 20 to page 4 line 3, shown in red font).